# Comparative Assessment of Gold and Carbon Nanoparticles as Tags for Lateral Flow Immunoassay of Fenpropathrin in Green Tea

**DOI:** 10.3390/foods14162806

**Published:** 2025-08-13

**Authors:** Chen Chen, Jinglei Xia, Jing Wang, Hongxing Wei, Qianxin Liang, Ziye Feng, Huimei Cai, Qingkui Fang, Ruyan Hou, Hongfang Li

**Affiliations:** 1Anhui Provincial Key Laboratory of Food Safety Monitoring and Quality Control, Joint Research Center for Food Nutrition and Health of IHM, Animal-Derived Food Safety Innovation Team of Anhui Agricultural University, College of Food and Nutrition, Anhui Agricultural University, Hefei 230036, China; 22721096@stu.ahau.edu.cn (C.C.); 15956985772@163.com (J.X.); wj2323358723@126.com (J.W.); whx546725331@163.com (H.W.); 22721093@stu.ahau.edu.cn (Q.L.); fzy1719@163.com (Z.F.); chm@ahau.edu.cn (H.C.); 2National Key Laboratory for Tea Plant Germplasm Innovation and Resource Utilization, School of Plant Protection, Anhui Agricultural University, Hefei 230036, China; 3Key Laboratory of Agri-products Quality and Biosafety, Ministry of Education, Anhui Agricultural University, Hefei 230036, China; qkfang@163.com

**Keywords:** lateral flow immunoassay, tea, fenpropathrin, gold nanoparticles, carbon nanoparticles

## Abstract

Fenpropathrin (FPT) residues in tea pose a threat to consumers’ health. Lateral flow immunoassay (LFIA) offers a rapid and convenient approach for FPT detection. However, existing LFIA formats, particularly those employing fluorescent nanoparticles, are susceptible to interference from the tea matrix’s endogenous fluorescence, limiting their accuracy for FPT analysis. Here, two complementary LFIAs based on gold nanoparticle (AuNP) and carbon nanoparticle (CNP) tags were developed, both of which avoid matrix fluorescence effects due to their intrinsic coloration under ambient light. We systematically evaluated five cleanup materials and identified polyvinylpolypyrrolidone as the optimal cleanup material. Following PVPP-based purification, green tea extracts required only a four-fold dilution prior to LFIA analysis, effectively minimizing matrix interferences, such as tea polyphenols, and enhancing assay robustness and sensitivity. The visual limit of detection was 0.64 μg/g and a quantitative limit of detection (qLOD) was 0.11 μg/g for FPT in green tea using the AuNP-based LFIA. The CNP-based LFIA further improved sensitivity, with a visual limit of detection of 0.16 μg/g and a qLOD of 0.017 μg/g, a 6.4-fold enhancement in qLOD compared to the AuNP assay. Together, these two assays provide an efficient and accurate strategy for on-site screening of FPT residues in green tea.

## 1. Introduction

Green tea is widely appreciated for its unique flavor and its abundance of health-promoting compounds, including tea polyphenols, amino acids, vitamins, etc. [1]. During the cultivation of tea plants, the use of pesticides is an effective strategy to control pest and weed infestations. Fenpropathrin (FPT), a pyrethroid pesticide, affects the gating kinetics of voltage-sensitive sodium channels, thereby blocking neural signaling and causing acute neurotoxicity in insects and non-target organisms [2]. This leads to hyperexcitation, paralysis, and even death of insects [3]. Due to its broad-spectrum efficacy, high effectiveness, and cost-efficiency, FPT is widely employed in tea plantations for insect control. However, incidents of FPT residues in tea, often resulting from excessive usage and non-compliance with prescribed withdrawal periods, are common. These residues can cause clinical symptoms, such as dizziness, headaches, and nausea, presenting a serious threat to consumer health [4]. To protect public health, many countries and organizations have set maximum residue limits for FPT in tea. In China, the maximum residue limit for FPT is set at 5 mg/kg in tea, according to the GB 2763-2021 standard [5]. In contrast, Japan and the European Union have set maximum residue limits of 25 mg/kg and 2 mg/kg in tea, respectively [6,7]. Consequently, a sensitive and rapid detection method is essential for monitoring FPT residues and ensuring consumer safety.

Traditional detection methods for FPT primarily rely on gas chromatography–tandem mass spectrometry (GC-MS/MS) [8]. However, these methods are limited by complex sample preparation, long detection times, and the requirement for professional operators. In contrast, rapid detection methods, such as lateral flow immunoassay (LFIA), enzyme-linked immunosorbent assay, and chemiluminescence, have been widely used in food analysis, environmental monitoring, and medical diagnosis [9,10]. Among these methods, LFIA combines immunological principles with chromatographic technology and has been successfully used in the manufacture of test strips. LFIA offers several advantages, including rapid results, simplicity, and cost-effectiveness [11]. At present, many companies, such as Abbott Laboratories, F. Hoffmann-La Roche AG, and Bio-Rad Laboratories, now produce mature LFIA test strip products. This shows that LFIA is indeed a method with great application prospects. Nanotags play a crucial role in LFIA, as they provide essential signal generation, enhancing both sensitivity and stability. While various nanotags have been reported, including gold nanoparticles (AuNPs), carbon nanoparticles (CNPs), silver nanoparticles, and quantum dots, only a few have been successfully developed into commercial test strips [12]. Quantum dots offer a higher signal-to-noise ratio, potentially improving assay sensitivity. For example, Zhang et al. synthesized double-layered quantum dot microspheres using silica for the detection of influenza viruses. Its visual sensitivity is eight times higher than that of traditional AuNPs-LFIA [13]. In addition, Cheng et al. modified long-range surface plasmon resonance with Au-nanoshells, which improved the sensitivity by 1.92 times compared with conventional long-range surface plasmon resonance, reaching 0.20 μg/mL [14]. However, their preparation involves a complex and costly process, including transitioning from the oil phase to water phase, and they contain heavy metals, which can cause significant environmental pollution [9]. Silver nanoparticles face issues with low stability due to their tendency to aggregate and oxidize, limiting their reliability in LFIA applications [15]. Thus, the shortcomings hinder their reliability and practical use in LFIA tests. In contrast, AuNPs are the most widely used nanotags, prized for their superior optical properties and straightforward synthesis [16]. CNPs offer a balance of performance and practicality: their low toxicity and high stability make them ideal for on-site testing [17]. A comparison of these nanoparticle types suggests that AuNPs and CNPs offer better application potential for LFIA development.

Currently, only a few LFIAs have been reported for the detection of FPT. One AuNP-based LFIA was developed to detect FPT in apples and cucumbers, with a limit of detection of 62 ± 6 μg/L [18]. Another AuNP-based LFIA was created for FPT detection in tea leaves, exhibiting a visual limit of detection (vLOD) of 5 μg/g [19]. Recent research on detecting FPT using LFIA still faces two key challenges: (1) A systematic comparison of AuNPs and CNPs as tags for FPT detection in tea—particularly under complex matrix interference—is lacking, and current approaches offer insufficient accuracy. (2) Existing cleanup methods for tea samples lack universality and efficiency. Substances in tea, such as polyphenols, caffeine, and pigments in tea, can cause significant matrix effects, which affect the detection performance of LFIA [20]. Polyphenols spontaneously bind to antibody proteins through hydrogen bonds and van der Waals forces, thereby covering the active sites on the antibody surface [21]. Meanwhile, caffeine, theaflavins, and thearubigins in the tea infusion can also affect antibody activity through hydrophobic interactions and non-specific adsorption. In addition, studies have confirmed that substances in tea, like catechin, gallic acid, and aspartic acid, can make the tea infusion acidic overall, causing slight deformation of antibody proteins and thus affecting their active sites. Substances, such as carotenoids and flavonoids, contained in green tea exhibit obvious green endogenous fluorescence at 500 nm, which interferes with fluorescent LFIA [22]. AuNPs and CNPs display intrinsic red and black coloration under ambient light, obviating the need for external UV excitation and thereby minimizing matrix interferences from tea. Consequently, there are relatively few LFIA focusing on pesticide residue detection in tea. Additionally, studies on LFIA in tea samples often lack an exploration of the specific impact of tea matrix components on detection performance. Therefore, it is crucial to investigate targeted removal of interfering tea matrix components to improve LFIA stability and propose a broadly applicable sample pretreatment method.

In this study, we established a new pretreatment method for LFIA to address the long-standing matrix effect issue in tea. We systematically compared, for the first time, the performance characteristics, such as limit of detection and specificity, when AuNPs and CNPs are used as signal tags. Meanwhile, we successfully developed AuNP-based and CNP-based LFIA with FPT as the target. The accuracy of these two LFIA methods was verified by instrumental analysis methods. These two methods can be cross-validated to improve detection precision, providing technical support for FPT detection in green tea.

## 2. Materials and Methods

### 2.1. Reagents and Instruments

The details of the reagents and instruments used in this study are provided in the Appendix A.

### 2.2. Synthesis of AuNPs and CNPs

AuNP solutions were prepared using the trisodium citrate reduction method as the previous studies, with some modifications [23]. Briefly, a clean 200 mL conical flask was filled with 0.2 mL of 5% chloroauric acid solution and 100 mL of ultrapure water, and then placed on a magnetic heating stirrer and heated to 60 °C. Next, 0.85 mL of 1% trisodium citrate solution was added and stirred with a magnetic stirrer (715 g). Subsequently, the mixture solution was heated for 10 min to obtain the AuNP solution, which was then allowed to cool naturally to room temperature. CNPs were prepared according to the previous studies [24]. The obtained CNPs (10 mg) were dissolved in a borate buffer solution (100 mL) to prepare the CNP solution.

### 2.3. Preparation of Immunoprobes

The pH of AuNP and CNP solution (1 mL) was adjusted with 0.1 M K_2_CO_3_, followed by the addition of FPT antibody (Ab) and incubation for 40 min. Then, the blank sites on the surface of the AuNP and CNP solutions were blocked with 10% bovine serum albumin and incubated for 20 min. Finally, the immunoprobes (AuNPs-Ab and CNPs-Ab) were obtained by centrifugation at 9168× *g* for 20 min to remove all supernatants. The immunoprobes were reconstituted in 100 µL storage solution.

### 2.4. Optimization of AuNP-Based LFIA and CNP-Based LFIA

The detection procedure was as follows: 10 μL of the immunoprobe was added to the microplates, and then 90 μL of sample solution (or FPT standard solution) was added and incubated for 3 min, followed by the insertion of the test strips. The detection results were visually observed 5 min later.

Several critical parameters were optimized. K_2_CO_3_ solution (0.1 M) at 30, 60, 90, 120, and 150 μL was added to the AuNP solution, and then the immunoprobes were prepared to evaluate the preferred pH value. Five diluent solutions that are frequently used in our lab were utilized to dilute antibody and coating antigen, e.g., water, phosphate-buffered saline (PBS, 0.1 mol/L, pH 7.4), 4-(2-hydroxyethyl)-1-piperazineethanesulfonic acid (Hepes, 0.1 mol/L, pH 7.4), (hydroxymethyl) methylaminomethane-HCl (Tris-HCl, 0.05 mol/L, pH 7.4), and phosphate (PB, 0.2 mol/L, pH 7.4). The concentrations of FPT antibodies were set as 0.12, 0.20, 0.28, 0.36, 0.44, and 0.52 μg/mL for AuNP-based LFIA, respectively. The concentrations of FPT antibodies for CNP-based LFIA were the same as those of AuNP-based LFIA, except for 0.12 μg/mL. The concentrations of coating antigen were set as 55, 33, 25, 20, and 15 μg/mL for AuNP-based LFIA, and then uniformly sprayed onto the nitrocellulose (NC) membrane. An extra concentration of coating antigen was set as 15 μg/mL for CNP-based LFIA. The AuNP-based immunoprobes were tested at the volumes of 4, 6, 8, 10, and 12 μL, while the CNP-based immunoprobes were evaluated at volumes of 1, 2, 4, 6, and 7 μL. When all conditions were optimized, the positive concentration was 1 μg/mL (in Tris-HCl buffer).

### 2.5. Pretreatment of Green Tea

The pretreatment of green tea samples was carried out with some modifications based on previous studies [19,25]. Green tea was homogenized to the powder (40 mesh). Then, green tea powder (1 g) was extracted with 80% methanol solution (4 mL) and vortexed for 2 min, followed by centrifugation for 5 min at 3300× *g*; this procedure was repeated for five replicates. The supernatant (1 mL) was collected, and then the 0.05 g of polyvinylpolypyrrolidone (PVPP), polyvinylpyrrolidone (PVP), sodium poly (styrene sulfonate) (PSS), graphitized carbon black (GCB), and polyethyleneimine (PEI) was separately added to explore the superior clean-up material and confirm the sample pretreatment method. After 5 min of purification, the mixture solution was centrifuged, and the supernatant (0.1 mL) was diluted 4-fold before LFIA analysis.

### 2.6. Analytical Performance of LFIA

In order to assess the analytical performance of LFIA, the sensitivity, specificity, accuracy, and stability of LFIA were analyzed. Briefly, the sensitivity of LFIA was analyzed by adding different concentrations of FPT to negative green tea to construct a standard calibration curve. The specificity of LFIA was determined by testing samples containing cycloprothrin, bifenthrin, isocarbophos, chlorpyrifos, thiamethoxam, fenitrothion, and carbofuran at the concentrations of 16 μg/g for AuNP-based LFIA, and 4 μg/g for CNP-based LFIA, respectively. The accuracy of LFIA was assessed by the spiked and recovery experiments and testing real green tea samples. Negative samples were spiked with FPT at 0.64, 1.0, 2.0, and 4.0 μg/g as well as detected using AuNP-based LFIA, and at 0.05, 0.10, 0.20, and 0.40 μg/g for the CNP-based LFIA. Meanwhile, 20 real green tea samples were also analyzed by AuNP-based and CNP-based LFIA, with all results corroborated by GC-MS/MS. The stability of LFIA was verified by comparing the color intensity of test strips stored at 37 °C for 1, 7, 14, 21, and 28 days.

### 2.7. Statistical Analysis

The results of this study are presented as the mean ± standard deviation (SD) of triplicate measurements (*n* = 3). The color intensity of the T lines was analyzed using Image J software (version 1.52p), with the quantitative data recorded as gray area values. Statistical analysis and data visualization were performed using Origin (version 2021).

## 3. Results and Discussion

### 3.1. Principle of AuNP-Based LFIA and CNP-Based LFIA Detection

The AuNP-based and CNP-based LFIA test strips share the same components (Figure 1): a polyvinyl chloride (PVC) backing plate, a sample pad, a nitrocellulose (NC) membrane, and an absorbent pad. On the NC membrane, the test line (T line) is coated with the FPT coating antigen, and the control line (C line) is coated with goat anti-mouse IgG.

If the sample is negative, the immunoprobe couples with the coating antigen and forms a distinct T line [26]. Thus, when the color intensity of the T line is markedly higher than that of the C line, the result is interpreted as negative. If FPT is present in the sample, it competes with the immobilized coating antigen for binding to the immunoprobe. Once the antigen-binding sites on the immunoprobe are occupied by FPT that is present in the sample, the immune complexes are unable to bind to the coating antigen on the T line. As the FPT concentration increases, the more antigen-binding sites on immunoprobes are occupied with FPT and cannot bind with the coating antigen, resulting in a progressively weaker T line signal. Hence, when the color intensity of the T line is equal to or lower than that of the C line, the result is interpreted as positive. Notably, the CNP-based LFIA operates on the same detection principle, differing only in its use of CNPs as the colorimetric label instead of AuNPs.

### 3.2. Characterization of AuNPs, CNPs, and Immunoprobes

As critical signal-labeling components in LFIA, the successful synthesis of AuNPs and CNPs coupled with their comprehensive characterization formed the fundamental basis for developing LFIA detection platforms. As shown in the inset of Figure 2a, the AuNP solution was burgundy-red, transparent, and stable, exhibiting no visible aggregation or precipitation. The transmission electron microscopy image in Figure 2a showed that the AuNP solution was ellipsoidal with a uniform size of about 40 nm. UV–Vis spectroscopy showed that the maximum absorbent peak of AuNPs was 529 nm (Figure 2b), and the hydration size of AuNPs was about 45 nm (Figure 2c). The integration of transmission electron microscopy, ultraviolet–visible spectroscopy, and hydration size analysis, as complementary characterization approaches, demonstrated that the AuNPs were synthesized successfully. This is consistent with the trend reported by Wang et al. [27]. The CNP solution exhibited a characteristic black color, which obscured distinct absorption peaks in UV–Vis spectrophotometric analysis and interfered with the accurate measurement of hydrodynamic size. Consequently, in this study, the CNPs were characterized by transmission electron microscopy, which was consistent with the literature reports [28,29]. As depicted in Figure 2d, the CNPs were well-defined nanoparticles with an average diameter of approximately 50 nm.

The successfully prepared AuNPs and CNPs were conjugated with the FPT monoclonal antibody (mAb FPT) to prepare immunoprobes. As shown in Figure 2c, compared with AuNPs, the hydrodynamic diameter of AuNPs-Ab increased significantly. Moreover, the ZETA potential of the immunoprobes increased compared with AuNPs and CNPs (Figure 2e,f), which indicated that the immunoprobes were successfully synthesized and could be used for the development of LFIA.

### 3.3. Development of AuNP-Based LFIA and CNP-Based LFIA

#### 3.3.1. Optimization of the Key Parameters for LFIA

To improve the sensitivity of AuNP-based and CNP-based LFIA, the key parameters were optimized. In this study, when the color intensity of the T line exceeds 6000, a higher inhibition rate indicates greater sensitivity of the method, and the corresponding conditions at this time are the optimal conditions. We first examined the effect of K_2_CO_3_ volume for AuNP-based LFIA. As shown in Appendix A, increasing the volume of K_2_CO_3_ from 30 to 90 µL led to a decline in color intensity of T line in the presence or absence of FPT even as the inhibition rate increased. At 150 µL of K_2_CO_3_, the inhibition rate reached its maximum, but the color intensity on the T line became weak, potentially increasing false positives. Hence, the K_2_CO_3_ at 90 µL, with the medium gray area and the higher inhibition ratio (58.4%) was chosen as the optimal volume. Because different types of antibody diluents can affect the labeling efficiency of antibodies on the surface of AuNPs or CNPs, we optimized the antibody diluent. It is obvious that when the diluent solution of the antibody was PBS, the inhibition was the highest, and the color intensity met the analytical requirements (Figure 3a,b). Precisely balanced concentrations for antibody and coating antigen promote the rapid and efficient formation of immune complexes at the T line, maximizing signal strength while minimizing background noise and non-specific interactions. Here, the antibody concentration was then evaluated. As shown in Figure 3c,d, with the increase in antibody concentration, the T line color intensity deepened progressively, both in the presence and absence of the FPT, while the inhibition ratio initially declined, plateaued, and then declined further. Although inhibition ratios (69.7 and 69.8%) were relatively high when antibody concentration reached 0.12 and 0.20 µg/mL, the corresponding color intensity of the T line (negative control) remained low. When the antibody concentrations were at 0.28 and 0.36 µg/mL, inhibition ratios were comparable. But the higher and more stable color intensity in T line was observed when the antibody concentration was at 0.36 µg/mL, which indicated superior performance. Different diluent solutions for coating antigen influence the binding interaction between the immunoprobe and the coating antigen. Here, we found that both the inhibition ratio (77.1%) and color intensity were greater when the coating antigen was diluted with PB (Figure 3e,f).

As the concentration of coating antigen increased, the color intensity for T line gradually faded both in the presence and absence of the FPT, and the inhibition ratio first increased and then decreased. The highest inhibition rate (57.5%) was observed at a coating antigen concentration of 25 µg/mL (Figure 3g,h). The greater the volume of the immunoprobe, the deeper color intensity of the T line; however, an excessive amount of immunoprobes may negatively affect sensitivity. Therefore, we explored the volume of the immunoprobes. As shown in Appendix A, the intensity of color of the T line gradually increased with an increasing volume of immunoprobes, but the inhibition ratio peaked (48.3%) at a volume of 8 µL and then gradually decreased. Therefore, in the AuNP-based LFIA, the optimal dilutions for the antibody and coating antigen were PBS and PB, respectively, with optimal concentrations of 0.36 µg/mL and 25 µg/mL, and the optimal volume of the immunoprobe was 8 µL.

We employed the same evaluation strategy used for the AuNP-based LFIA to identify the optimal conditions for the CNP-based LFIA. Different diluent solutions for the antibody significantly influence the color intensity of T line and the inhibition ratio. As shown in Figure 3i,j, the inhibition rate (74.8%) was highest, and the color intensity was also greater when the antibody dilution was PB. At an antibody concentration of 0.28 µg/mL, the highest inhibition ratio was observed (Figure 3k,l). As shown in Figure 3m–p, the highest inhibition ratio and greater color intensity were achieved when the coating antigen was diluted in PBS at a concentration of 25 µg/mL. When the volume was 4 μL, the inhibition ratio reached a peak, and the color intensity of the T line was relatively high, which was sufficient for LFIA (Appendix A). Therefore, the optimal antibody and coating antigen dilutions were PB and PBS, and the optimal concentrations of the antibody and coating antigen were 0.28 µg/mL and 25 µg/mL for the CNP-based LFIA.

#### 3.3.2. Development of AuNP-Based and CNP-Based LFIA in Buffer Solution

Under the optimal conditions, standard curves for FPT were constructed using AuNP-based and CNP-based LFIA. As the concentration of the FPT increased, the color intensity of T line gradually declined. Due to insufficient antibody sensitivity, the AuNP-based LFIA still exhibited a visible T line at 1 μg/mL (Figure 4a,b). In contrast, the CNP-based LFIA, owing to its higher sensitivity, showed a virtually complete disappearance of the T line at 0.5 μg/mL (Figure 4e,f). In this research, we determined the qualitative detection limit of the LFIA using visual colorimetric interpretation. When the color intensity of the T line exceeds that of the C line, the detection result is considered negative. The vLOD is defined as the minimum FPT concentration at which the color of the T line becomes noticeably lighter [30,31]. Hence, the vLOD were 0.04 and 0.01 μg/mL for AuNP-based LFIA and CNP-based LFIA in buffer solution. Subsequently, the color intensity of the T lines was recorded and fitted using the four-parameter equation, and the R^2^ values were both higher than 0.994 [32].

#### 3.3.3. Sample Pretreatment Confirmation of Green Tea

Green tea is one of the most widely consumed teas worldwide [33]. It is rich in polyphenols, catechins, caffeine, amino acids, etc., that may adversely affect the performance of LFIA. Numerous studies have indicated that PVPP, PVP, PSS, PEI, and GCB, respectively, exhibit good adsorption properties for substances with matrix effects such as polyphenols, metal ions, and pigments [21,34,35,36,37]. In this study, we employed these five clean-up materials commonly used in instrumental analysis to adsorb tea matrix components, and then evaluated the effectiveness of each material using the AuNP-based LFIA. As shown in Appendix A, when PEI was used as the clean-up material, its excessive viscosity prevented effective separation of the purified green tea extract; therefore, PEI was excluded from subsequent experiments. Then, the green tea sample solution was centrifuged, diluted four-fold (Appendix A), and the supernatant was applied to the test strips to evaluate the purification performance of the remaining reagents. As shown in Appendix A, when PSS was used as the clean-up material, the T line was completely absent, indicating that PSS reduces the affinity between the coating antigen and the immunoprobe. When GCB was used, the NC membrane exhibited virtually no background coloration, confirming GCB’s ability to adsorb chlorophyll; however, its inhibition ratio was still lower than that in the buffer solution. Both the unpurified sample and the PVP-treated sample showed significantly reduced inhibition ratios compared to the buffer control. Only PVPP as the clean-up material achieved the highest inhibition ratio. As reported, PVPP and PVP both interact with polyphenolic compounds, but their physicochemical differences lead to distinct behaviors in sample pretreatment. PVPP is a cross-linked, insoluble polymer. When added to an aqueous extract, it rapidly adsorbs phenolic interferents and can be removed by simple centrifugation or filtration, leaving a clear supernatant for analysis [38]. PVP, in contrast, is fully soluble. Although it also binds phenolics, it remains in solution and cannot be removed by phase separation [39]. We hypothesize that PVP dissolved in the green tea extract diminishes the affinity between the coating antigen and the immunoprobe. Accordingly, PVPP exhibited significantly superior purification performance compared to PVP and was thus selected as the purification agent for green tea in LFIA development.

To further evaluate whether the established sample pretreatment is suitable for both AuNP-based and CNP-based LFIA, we constructed matrix standard curves. The color intensity trends of the T line in the test strips and standard curve profiles closely paralleled those obtained in the buffer system, confirming that this pretreatment effectively extracts FPT from green tea for LFIA analysis (Figure 4c,d,g,h).

### 3.4. Detection Performance of AuNP-Based and CNP-Based LFIA

The relationship between the concentration of the analyte and the signal is sigmoidal in immunoassays. IC_10_ (10% inhibitory concentration) refers to the concentration of the target analyte that reduces the signal intensity by 10% in a competitive immunoassay system. Thus, the quantitative limit of detection (qLOD) of FPT was defined as the IC_10_ derived from the standard curve [40,41]. The vLOD were 0.64 μg/g and 0.16 μg/g for AuNP-based and CNP-based LFIA, respectively (Figure 4c,g). The quantitative limits of detection (qLOD) of AuNP-based and CNP-based LFIA were 0.11 μg/g and 0.017 μg/g in green tea, with linear ranges of 0.31–4.93 μg/g and 0.03–0.55 μg/g, respectively. The vLOD and qLOD of CNP-based LFIA were improved by 4-fold and 6.4-fold, respectively, in comparison to AuNP-based LFIA.

We compared the proposed AuNP and CNP-based LFIA with the reported rapid detection methods in detecting pyrethroid pesticides (Appendix A). Compared with fluorescent immunoassay and fluorescent sensors, LFIA is the most commonly employed method. Although fluorescent nanomaterials exhibit high sensitivity, their application is limited by issues, such as pH sensitivity and self-quenching phenomena, and they all have the limitation of a relatively long detection time (15–60 min) [42,43,44]. In comparison, LFIA offers shorter detection times (10–30 min) and is suitable for point-of-care detection [18,19]. Gao et al. reported an AuNP-based LFIA for determining FPT residues in tea [18]. However, they eliminated the matrix effect by diluting the tea sample solution 50-fold, which reduces the method’s sensitivity. In our study, we systematically optimized the tea sample pretreatment, and the purified tea sample solution only needs to be diluted four-fold for direct detection.

Traditional AuNPs are commonly used as colorimetric probes in LFIA, but their sensitivity is relatively low. In contrast, this study successfully developed a LFIA method based on AuNP and CNP probes, demonstrating that CNPs offer superior sensitivity and stability compared to conventional AuNPs. In addition, there are relatively few LFIA methods for detecting FPT residues in tea. This study not only successfully established a LFIA using CNPs, which improved the sensitivity, but also filled the gap in the lack of new material alternatives in this detection field. AuNPs offer easier conjugation and more intuitive red signals, making them suitable for rapid on-site screening. CNPs can not only be conjugated with antibodies through electrostatic adsorption, but also have characteristics such as higher sensitivity and lower cost. Meanwhile, they are environmentally friendly materials. Both the black signal of CNPs and the red signal of AuNPs have obvious color differences from the white backplane, resulting in good signal reading performance. In addition, both CNPs and AuNPs have good stability. In general, CNPs have the characteristics of lower cost and higher sensitivity, thus having broad application prospects. Furthermore, the limits of detection achieved in this study are substantially lower than established maximum residue limits, and the dual-platform (AuNP- and CNP-based) formats enable mutual cross-validation. This dual-validation strategy further enhances method robustness, rendering it a reliable and sensitive approach for quantifying FPT residues in tea.

The specificity of the proposed LFIA was assessed by measuring the cross-reactivity against seven non-target pesticides (Appendix A). The cross-reactivity for AuNP-based and CNP-based LFIA were both negligible (Both are 4%), demonstrating that the proposed LFIA is specific for FPT (Figure 4i,j). Recoveries for the AuNP-based LFIA ranged from 87.4 to 106.5% (RSD 6.44–11.75%), while those for the CNP-based LFIA ranged from 89.7 to 105.6% with RSD lower than 12.84% (Appendix A). We found that the color intensity of the T line exhibited no significant changes over the storage period, demonstrating that the test strips remain stable at 37 °C for at least 28 days (Appendix A). Arrhenius equation (K = A×e^-Ea/RT^) predicted that the test strips remain stable for at least 4 months at room temperature (25 °C) and for up to three years when stored at 4 °C. These results confirm the high long-term stability of both AuNP-based and CNP-based LFIAs, rendering them well suited for FPT monitoring in green tea samples. As shown in Figure 5a,b, the T line color intensity for all 20 green tea samples were identical to the negative control, indicating that every sample tested negative. Furthermore, GC-MS/MS analysis also confirmed that all samples were negative (Appendix A). These findings demonstrate the excellent accuracy of the proposed LFIA.

## 4. Conclusions

In this study, the FPT monoclonal antibody was conjugated with AuNPs to construct an AuNP-based LFIA, which provides a reliable basis for on-site screening. To enhance sensitivity and stability, CNPs were further employed as alternative tags. Through systematic optimization of key parameters and methodological validation, a CNP-based LFIA with superior performance was successfully established. The vLOD and qLOD of the CNP-based LFIA were 0.16 μg/g and 0.017 μg/g, respectively, which were 4-fold and 6.4-fold lower than those of the AuNP-based LFIA (vLOD: 0.64 μg/g; qLOD: 0.11 μg/g). Additionally, we found that PVPP was the optimal cleanup material for green tea in the detection of FPT using LFIAs, providing a practical strategy for the rapid purification of green tea samples. The proposed AuNP-based and CNP-based LFIA, when combined with the sample pretreatment method, exhibited good accuracy, stability, and specificity, showing broad application prospects for on-site monitoring and risk assessment of pesticide residues in complex tea matrices. However, this study focuses on green tea, and its applicability to other tea types (such as black tea and oolong tea) needs further verification due to their different matrix compositions.

## Figures and Tables

**Figure 1 foods-14-02806-f001:**
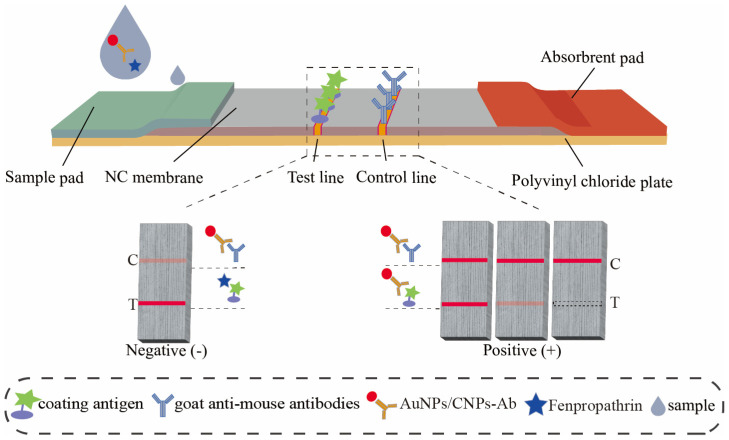
Schematic illustration of the detection principle of AuNP/CNP-based LFIA.

**Figure 2 foods-14-02806-f002:**
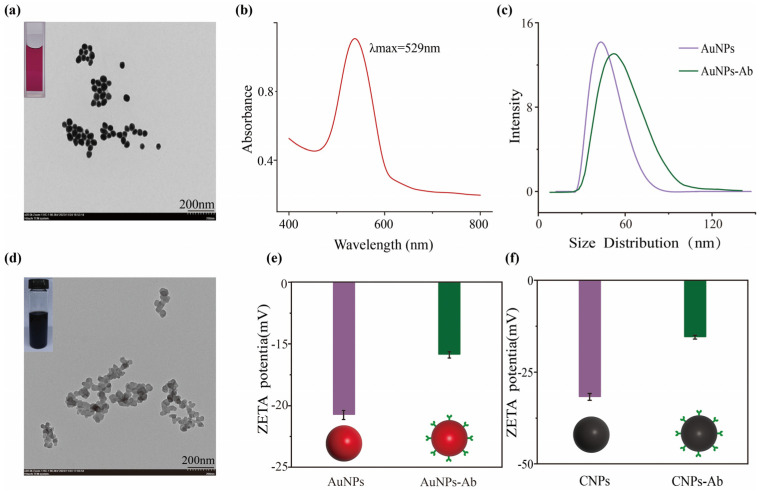
TEM (**a**) and UV–Vis (**b**) of AuNPs. UV–Vis (**c**) and zeta potential (**e**) of AuNPs and AuNPs-Ab. TEM (**d**) of CNPs and zeta potential (**f**) of CNPs and CNPs-Ab. Each test was repeated three times (*n* = 3).

**Figure 3 foods-14-02806-f003:**
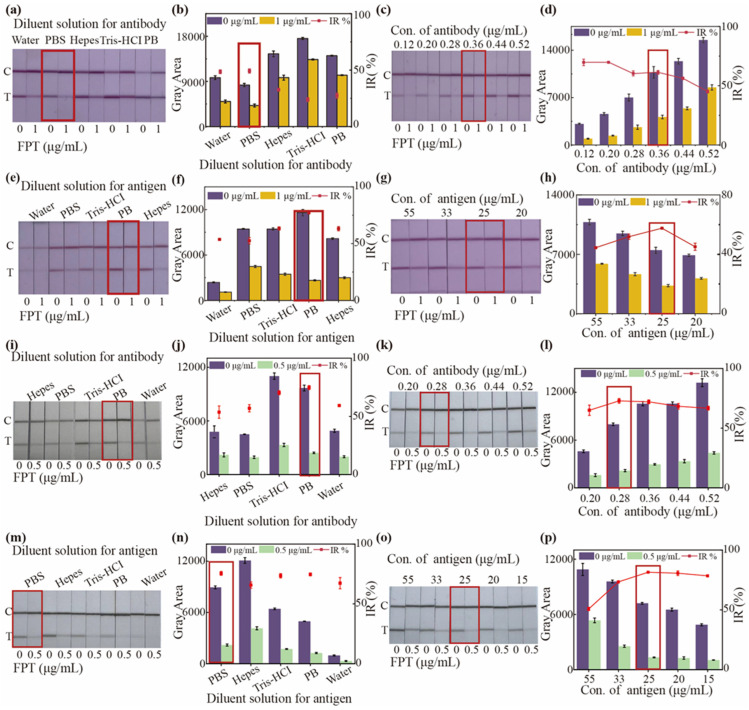
The visual image of test strips and the gray area of T line and inhibition ratio for optimizing (**a**,**b**) diluent solution for antibody, (**c**,**d**) antibody concentration, (**e**,**f**) diluent solution for antibody, and (**g**,**h**) antigen concentration for AuNP-based LFIA. The visual image of test strips and the gray area of T line and inhibition ratio for optimizing (**i**,**j**) diluent solution for antibody, (**k**,**l**) antibody concentration, (**m**,**n**) diluent solution for antibody, and (**o**,**p**) antigen concentration for CNP-based LFIA. IR and Con. were the abbreviations for inhibition ratio and concentration. The parameters marked with a red dotted frame were selected as the optimal conditions. Each test was repeated three times (*n* = 3). The color intensity was recorded with the gray area.

**Figure 4 foods-14-02806-f004:**
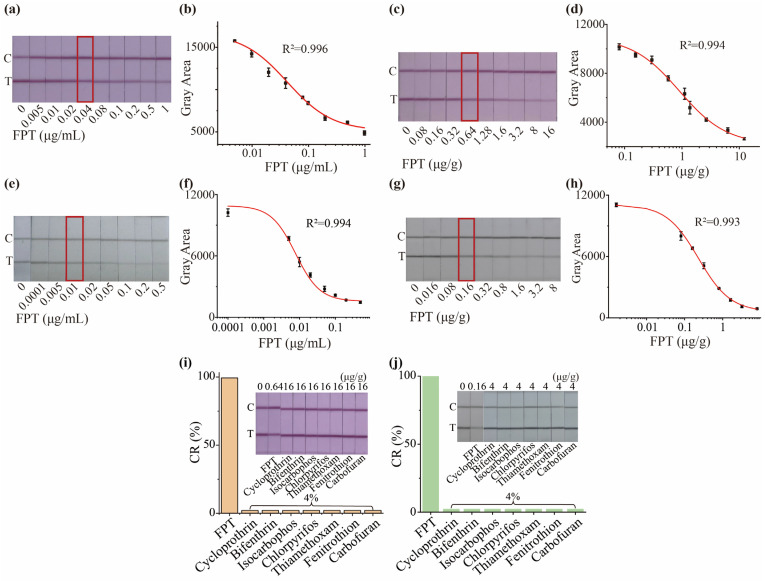
(**a**) The visual image of test strips and (**b**) standard curve for FPT using AuNP-based LFIA in buffer solution. (**c**) The visual image of test strips and (**d**) standard curve for FPT using AuNP-based LFIA in green tea. (**e**) The visual image of test strips and (**f**) standard curve for FPT using CNP-based LFIA in buffer solution. (**g**) The visual image of test strips and (**h**) standard curve for FPT using CNP-based LFIA in green tea. The specificity of (**i**) AuNP-based LFIA and (**j**) CNP-based LFIA.

**Figure 5 foods-14-02806-f005:**
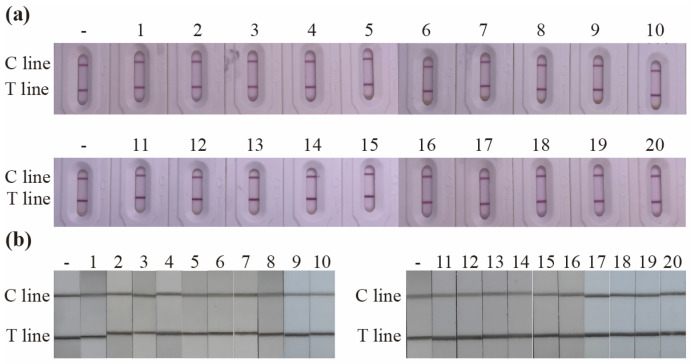
Real sample detection of AuNP-based LFIA (**a**) and CNP-based LFIA (**b**). Each test was repeated three times (*n* = 3). “-” means negative control.

## Data Availability

The original contributions presented in the study are included in the article/Appendix A, further inquiries can be directed to the corresponding author.

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
