# Peer review of "Comparative Assessment of Gold and Carbon Nanoparticles as Tags for Lateral Flow Immunoassay of Fenpropathrin in Green Tea"

_foods, 2025, doi:10.3390/foods14162806_

Round 1
Reviewer 1 Report
Comments and Suggestions for Authors
The study is methodologically sound, addresses a relevant food safety concern, and demonstrates innovation through dual LFIA design. However, the manuscript requires revision to improve scientific clarity, correct terminology, ensure consistency in methodological reporting, and enhance the rigor of data interpretation.
Introduction:
Describe in general green tea and its properties. Refer to https://doi.org/10.1016/j.chemosphere.2024.143550
Expand discussion of matrix effects specific to tea: polyphenols, caffeine, pigments
Focus more narrowly on gaps in current LFIA methods for fenpropathrin in tea
Clearly state the novelty
Materials and Methods:
The paragraph ‘Pretreatment of green tea’ lacks full rationale for the choice of each clean-up material
Number of replicates and statistical analysis are not clearly stated in this part
Results and Discussion:
Discuss trade-offs between sensitivity and signal intensity in choosing final parameters
Conclusions:
Indicate limitations of the study
Author Response
Thank you very much for your comments and suggestions. We have carefully revised the manuscript accordingly. Detailed responses to review comments are attached below. The comments were valuable and helped improve the manuscript. We hope that our correction of the manuscript could meet the requirement of acceptance. The revised parts are shown in red font in the manuscript.

Reviewer 2 Report
Comments and Suggestions for Authors
This manuscript presents the development and comparison of two lateral flow immunoassays (LFIA) using gold and carbon nanoparticles as labels for the rapid and accurate detection of fenpropathrin residues in green tea, with optimized sample cleanup using PVPP to reduce matrix interferences. While the manuscript is generally interesting and holds sufficient interest for Foods, it has several limitations. I recommend a major revision of the text.
Title: The syntax “Comparative Lateral Flow Immunoassays… Using Gold and Carbon Nanoparticles as Labels” could be streamlined, and the term “labels” might be replaced with “tags” or “reporter particles” for clarity.
Abstract:
Line 22: The claim that both AuNPs and CNPs “exhibit strong surface plasmon resonance signals” is probably scientifically incorrect. Gold NPs exhibit a plasmonic peak, but does carbon nanoparticles possess a plasmon resonance? This misleading statement should be corrected.
Lines 19–20: The abstract asserts that fluorescent-label LFIA are “susceptible to interference from the tea matrix’s endogenous fluorescence” without providing any citation or evidence. Such a claim requires supporting data or a reference, as it is not universally accepted that tea autofluorescence significantly confounds fluorescent LFIA.
Line 21: Parts of the manuscript use passive (e.g., “was prepared”, “was constructed”). However, phrases like “we developed” in the abstract are active. Since this is a report, “were developed” would fit the passive style. The authors should ensure consistency of voice.
Line 25: The Abstract states that a four-fold dilution of tea extract was used to minimize matrix effects, but neither the introduction nor methods give a rationale for choosing 4x specifically. Was this based on preliminary tests? The authors should explain how this factor was determined. A 4x dilution is relatively low for complex tea extracts; justification is needed.
Line 26: Phrase “enhancing assay stability” is vague. It presumably means reduced matrix effects or consistent signals, but “stability” often refers to shelf-life. The wording should be more precise (e.g. “robustness” or “reproducibility in matrix”).
Introduction
Scope and Novelty: The introduction is comprehensive, but the novelty claim is weak. Using AuNP vs CNP as LFIA labels is not entirely new (authors cite relevant refs), and the advantage of CNP here (6.4× qLOD) should be framed relative to other published improvements (e.g. Au-nanoshells, quantum dots). More recent references should be added to position this work in the current literature.
Material and methods:
Lines 287-293: The definition of the qualitative (visual) LOD is nonstandard. The manuscript defines vLOD as “the lowest FPT concentration for which the T-line intensity equals or falls below the C-line”. This is a subjective criterion (dependent on visual comparison of intensities) and not the usual statistical LOD (e.g. 3σ of blank). Moreover, the use of C-line intensity as a threshold should be justified. At minimum, the authors should reference a precedent or explain why this method is valid.
Results and discussion
The Results and Discussion section includes a limited number of references, and the manuscript would benefit from a more comprehensive comparison with findings from other relevant studies to better contextualize the results and highlight the novelty of the work.
Lines (176–177):–The Results text redundantly repeats the “absence of FPT” scenario in back-to-back sentences. Specifically, lines 172–173 repeat nearly the same description as lines 176–177. This duplication appears to be an editing oversight and should be removed for conciseness.
Line 176: There is a grammar issue: the sentence “If FPT is absent in sample, immunoprobe combines with coating antigen…” uses “combines” incorrectly. It should be “binds” or “couples” in this context. This verb misuse should be corrected.
Lines (178-187): The description of positive/negative interpretation is confusing. The text states that absence of fenpropathrin gives a strong T-line (negative result) and presence yields a weak T-line (positive result). This is counterintuitive (generally a “positive” test indicates presence of analyte), and the phrasing should be clarified or reworded to avoid reader confusion about which condition is considered a positive test.
Figure 1: The text refers to Figure 1a, Figure 1b, and Figure 1c, while the figure itself does not contain the labels a, b, and c. The figure panels should be correctly labeled.
L258: In describing the antibody/coating antigen optimization, the text incorrectly states that an “antibody concentration of 25 μg/mL” gave the highest inhibition. However, Figure 2g–h actually vary the coating-antigen concentration, not the antibody. The text should be corrected to say “coating antigen” instead of “antibody” (and ensure the correct concentration units: μg/mL refers to antigen).
Lines (297–299): The statement “PVPP and PVP could adsorb polyphenolics; PSS and PEI could adsorb metal ions; GCB adsorbs pigments” is supported by references [27–29,22]. However, reference 22 is Posthuma-Trumpie 2011 (on carbon NP labels), which is irrelevant here. References 27–29 likewise do not specifically discuss all these interactions in tea. The citations seem mismatched and do not clearly support the claimed cleanup functions. Each claim should be backed by a specific reference, or this sentence should be qualified/removed.
Line 229, Line 234: The Results text reports exact grayscale intensities. These raw values are instrument-dependent and add little interpretive value; they clutter the narrative. It would be better to report relative changes or percentages, rather than unverifiable raw numbers. As written, they distract from the conclusions without improving clarity.
Line 366: A supplemental table showing cross-reactivity with major analogs should be included.
The manuscript shows CNPs outperform AuNPs in LOD, but does not discuss trade-offs (e.g. ease of conjugation, cost, signal readout difference between black vs red lines). A critical discussion comparing the two labels qualitatively is needed.
Conclusion
The conclusion would be stronger if it focused on the key finding.
References: The majority of references date before 2021.
Comments on the Quality of English Language
Manuscript should been 'spell checked' and 'grammar checked'.
Author Response

(The authors gave the same response as above.)

Round 2
Reviewer 1 Report
Comments and Suggestions for Authors
The Authors have improved the manuscript. I have no more comments.
Reviewer 2 Report
Comments and Suggestions for Authors
I have no further objections; I am satisfied with the author's response.